# Effect of silver diamine fluoride and proanthocyanidin on resistance of carious dentin to acid challenges

**Maryam Firouzmandi[1], Fateme Vasei[2]\*, Rashin Giti[3], Hadis Sadeghi[4]**

**1** Oral and Dental Disease Research Center, Department of Operative Dentistry, School of Dentistry, Shiraz University of Medical Sciences, Shiraz, Iran, **2** Department of Operative Dentistry, School of Dentistry, Shiraz University of Medical Sciences, Shiraz, Iran, **3** Department of Prosthodontics, School of Dentistry, Shiraz University of Medical Sciences, Shiraz, Iran, **4** Student Research Committee, School of Dentistry, Shiraz University of Medical Science, Shiraz, Iran

\* Fvj_anjell@yahoo.com

## Abstract

The aim of this study was to evaluate the effect of silver diamine fluoride and grape seed extract on the microstructure and mechanical properties of carious dentin following exposure to acidic challenge. Ninety-eight molars with occlusal caries were used. In the control group the specimens were kept in distilled water. In the GSE group, the specimens were immersed in 6.5% grape seed extract solution for 30 minutes. In the SDF group, the specimens were immersed in 30% SDF solution for 4 minutes. In the GSE+SDF group, the specimens were immersed in 6.5% grape seed extract solution for 30 minutes and then exposed to 30% SDF solution for 4 minutes. All the groups underwent pH cycling model for 8 days. Microhardness measurements were taken at the baseline before surface treatments and after pH cycling. Elastic modulus was measured, after pH cycling. In the control group, the final hardness was significantly lower than the initial hardness (P = 0.001). In the SDF group, the final hardness was significantly higher than the initial hardness (P < 0.001). There was no significant difference between the initial and final hardness values in the GSE and GSE + SDF groups (p = 0.92, p = 0.07). The $H_1$-$H_0$ in the SDF group was significantly higher than the other groups (P<0.05). Moreover, elastic modulus of the experimental groups except GSE+SDF group was significantly higher than control. The highest mean elastic modulus was detected in the SDF group (P<0.001). The use of SDF and GSE prior to the acid challenge improved mechanical properties. Microstructural investigation, using scanning electron microscope showed dentin structure protection against acid challenges with SDF treatment and collagen matrix stabilization with GSE treatment. However combined use of these agents was not beneficious.

## Introduction

Prevalence of dental caries as a preventable disease has decreased in recent decades. However it still remains as a prevalent and costly disease worldwide [1]. Current concepts in managing

**Data Availability Statement:** All relevant data are within the manuscript and uploaded files.

**Funding:** This study was supported by the vice chancellor for research of Shiraz University of Medical Sciences (Grant#8794123).

**Competing interests:** The authors have declared that no competing interests exist.

dental caries have shifted towards the repair of carious lesion rather than replacing it with restorative materials. Applying this concept to enamel lesions merely requires a chemical remineralization approach. However, dentin caries management is more challenging due to its complex biologic nature. One therapeutic modality proposed for dentin caries, is nonrestorative cavity control. This caries management technique is indicated in cavitated dentin carious lesions that can be made cleansable. For permanent teeth, this approach might be suitable for root surface caries [2]. It is intended to prevent further tooth tissue loss and achieve dentin caries arrest [3]. Currently several issues are to be clarified to guide the practitioners regarding nonrestorative cavity control. This includes additional control measures such as remineralizing or dentin biomodifying agents. Hydroxyapatite and collagen in dentin structure are interdependent. Any attempt to biomodify carious dentin instead of the traditional surgical methods should be able to restore both components [4]. Dentin caries starts with mineral dissolution caused by acidic pH of the cariogenic plaque. During the next stage, inorganic component, mainly collagen matrix gets involved. Hydroxyapatite crystals surround collagen fibers and protect them. Exposed collagen network after dentin demineralization is prone to degeneration. Exposure of cleavage sites of collagen fibers after demineralization is the beginning of organic matrix degradation process [5]. In the past it was believed that bacterial proteases are responsible for dentin matrix degradation, but it was shown that these proteases are inactivated by acidic pH. Nonetheless, host-derived proteolytic enzymes, such as matrix metalloproteases (MMPs) and cysteine cathepsins are responsible for dentin collagen degradation [6]. It was reported that the presence of an organic matrix might diminish the progress of dentin erosion [7]. The preservation and stability of dentin collagen is essential during the remineralization process, since it acts as a scaffold for mineral deposition. In order to repair carious dentin, biological changes should first be made in the dentin matrix, and subsequently the mineral content should be replaced. Also, the repaired dentin should be able to withstand the acidic challenges of the oral cavity [8]. Accordingly, teeth will become more resistant to further demineralization.

Grape seed extract (GSE) contains proanthocyanidin (PA), a natural collagen cross-linker with polyphenolic compounds. Previous studies showed that plant-derived PAs are strong dentin biomodifying agents [9]. This agent was reported to strengthen collagen-based tissues by increasing collagen cross-links [10]. Also, it inhibits proteolysis of collagen molecule through bonding to the cleavage sites [11]. PA is known as non-specific inhibitor of MMPs, and its efficacy in deactivating cysteine cathepsins was previously shown [12]. Silver diamine fluoride (SDF) is an affordable, effective, safe, and easy to use caries arresting agent [13]. Clinical trials have exhibited promising demineralization inhibitory results after topical application of SDF solution [14–16]. Also, SDF can inhibit cariogenic bacteria growth in the biofilm [17]. Silver compounds penetrate into the dentinal tubules invading cariogenic microorganisms [13]. In addition, it can harden carious lesion [8, 16, 18], inhibit degradation of collagen in demineralized dentin [19], and exert an inhibitory effect on MMPs [6] and cysteine cathepsins [20]. Arresting coronal and root dentin caries [14, 18, 21, 22] by modifying both mineral and organic structure of dentin is the endpoint that makes SDF superior to other remineralizing agents.

Our previous research focused on the effect of PA and SDF on mechanical properties of carious dentin [8]. Although the immediate effect of SDF and SDF+PA on restoring mechanical properties of carious dentin was recorded, the resistance to further acid challenges was not investigated. Therefore, the present study was designed to investigate the effects of silver diamine fluoride and grape seed extract on the hardness and elastic modulus of carious dentin as well as its microstructural characteristics after exposure to acidic challenge. The null hypothesis was that treating the carious dentin with SDF, GSE or GSE+SDF before acid

challenge exerts no significant changes on the microstructure, hardness, and elastic modulus compared to the control.

## Materials and methods

Ninety-eight third molars with occlusal caries without any previous restorations extracted for surgical reasons were used following obtaining informed consent from patients under a protocol approved by the local Ethics committee of Shiraz University of Medical Science with the ethic number of IR.SUMS.REC.1394.S894. The specimens were kept in distilled water at 4˚C and used within one month after extraction.

### Sample preparation

International Caries Detection and Assessment System (ICDAS) was considered to select the teeth with caries extending into the outer third of dentin (ICDAS code 2) [23]. Occlusal enamel was grounded, using diamond disks (Isomet Low Speed Saw; BuehlerLtd, Lake Bluff, IL, USA) with water coolant to expose carious dentin. Caries Detector dye, tactile and visual examination were used to detect carious dentin. The carious area was examined with explorer to make sure of the softness. Caries Detector (Kuraray Medica Inc., Tokyo, Japan) was applied to the cut surface of the teeth, using a microbrush. After 2 minutes, the tooth surface was washed with water and the pink stained area was considered as being carious.

Forty teeth were prepared for the hardness test. A horizontal section was made through the CEJ area to remove the roots. Then, silicon carbide papers of 600–1200 grit (Sumare, SP, Brazil) were used to polish the specimens. A layer of nail varnish was applied on the surface, except a 5×5 mm area on the center of carious dentin lesion to localize the target area during the test. Then, the specimens were mounted in self-cure acrylic blocks.

Fifty teeth were used to determine elastic modulus in tensile test. In order to prepare the specimens, carious dentin was determined as described in the previous paragraph. Then, a horizontal cut was made using a water-cooled saw to prepare the dentin slabs with $0.5 \pm 0.1$ mm thickness. The dentin slabs were prepared to hourglass-shaped specimens, using an ultra-fine diamond bur mounted in a water-cooled high-speed handpiece. The test site was $0.5 \pm 0.1$ mm in width and $0.5 \pm 0.1$ mm in thickness, with a cross-sectional area of approximately 0.25 mm$^2$. Each slab was covered with a layer of nail varnish, except on the test site.

For SEM analysis the remaining eight teeth were prepared in a similar fashion to hardness, but specimens were cut from the center of the target area into two equal segments defined as the test or control. The test segments underwent surface treatments and acidic challenge according to the following procedure. Control segments were stored in distilled water at 4˚C during the study period.

### Experimental treatments

The specimens of microhardness test were divided into 4 equal groups as follows:

- Control: The specimens were kept in distilled water for 30 minutes at room temperature.

- GSE: 6.5 g powder of grape seed extract was collected from capsules containing this powder (Puritans Pride Inc., Oakdale, NY, USA) and dissolved in 100 ml distilled water [24]. The primary solution pH was recorded as pH = 5. To adjust it, the basic buffer of NaOH was added to the solution, which resulted in a neutral pH of 7. Then, the specimens were treated with the 6.5% grape seed extract for 30 minutes at room temperature and then rinsed with distilled water [25].

- SDF: The specimens were treated with 30% silver diamine fluoride solution (Ancarie, Cariostatico, Maquira Dental Products, Maringa, Brazil) for 4 minutes at room temperature and then rinsed with distilled water [26].

- GSE + SDF: At first, the specimens were treated with grape seed extract for 30 minutes, and then the SDF solution was applied for 4 minutes and washed with distilled water. Next, all the specimens were subjected to acidic challenge for eight days at 37˚C. The challenge involved six consecutive cycles each day. In each cycle, the specimens were immersed in the demineralization solution (1.5mmol/L $CaCl_2$, 0.9 mmol/L $KH_2PO_4$, 50 mmol/L acetate) for 30 minutes at pH = 5 and then washed with distilled water. Afterwards, the specimens were immersed in the remineralization solution (20 mmol/L 4- (2-hydroxyethyl) -1-piperazine ethanesulfonic acid (HEPES), 1.5 mmol/L $CaCl_2$, 0.9 mmol/L $KH_2PO_4$, 150 mmol/L KCl) for 10 minutes at pH = 7 and then rinsed with distilled water. The specimens were kept in natural buffer overnight [27].

For the elastic modulus measurement, the specimens were divided into 5 groups including the four treatment groups plus the carious dentin (CD) group (baseline).

## Mechanical tests

In this study, the surface hardness of the specimens was measured in two stages. In the first stage, the initial hardness ($H_0$) was measured prior to surface treatments and acidic challenge, which was considered as the baseline hardness. In the second stage, hardness was recorded after surface treatments and acid challenge and regarded as final hardness ($H_1$). Knoop microhardness (KHN) of the carious dentin was measured in a moist state immediately after removing from distilled water. The specimens were placed under the Knoop indenter of a microhardness tester (SCTMC, MHV-1000Z, China) and subjected to a load of 25 (kg/mm$^2$) for 20 s at each test point. Three measurements were made for each sample and the mean value was established as the KHN.

In order to measure elastic modulus, the trimmed specimens were glued to a tensile testing jig by means of cyanoacrylate adhesive (Zapit, Dental Ventures of America, Corona, California, United States). The jig was pulled at a rate of 0.6 mm/min in a universal testing machine (Instron; Zwich, Germany), and the modulus of elasticity was recorded in MPa.

## Surface morphology

To fix the test and control segments, all specimens were immersed in 4% formaldehyde solution for 24 hours. Then, they were washed in ultrasonic cleaner and dried by increasing the ethanol concentration (50%, 60%, 80% and 100%). Finally, the fixed and dried specimens were gold plated and examined, using SEM (Tescan Vega II, England). The relative amount of the depositions in the study groups were evaluated and presented in a histogram (Fig 7)

## Statistical analysis

The data were analyzed using SPSS, version 18 (SPSS Inc., Chicago, IL, USA). One-way ANOVA was used to compare hardness between the study groups, and Paired t-test was performed to show the significant differences in initial and final hardness. The data of elastic modulus were analyzed using one-way ANOVA and tukey HSD multiple comparisons. The level of significance was set at 0.05.

## Results

The results of mechanical tests are presented in Table 1. Study groups, in contrast to the control group showed an increase in H1 compared to H0. The lowest difference between H1 and H0 was observed in GSE+SDF group, and the highest was observed in SDF group (Fig 1). One-way ANOVA revealed that there was no significant difference in the baseline hardness between the groups (p = 0.12). The comparison of the H0 and H1 of each group, using t-test showed that in the control group, H1 was significantly lower than the H0 (p = 0.001). In the SDF group, H1 was significantly higher than the H0 (p < 0.001). There was no difference between the initial and final hardness of the GSE and GSE+SDF groups (p = 0.07 and, p = 0.92). One-way ANOVA also showed that H1-H0 was significantly different between the study groups (p<0.001). Further analysis, using tukey test indicated that the H1-H0 value of SDF group was significantly different from the other groups (p<0.001). GSE and GSE+SDF groups were not significantly different in H1-H0 values (p = 0.54). The H1-H0 of the control group was significantly lower than that of GSE group (p = 0.01) while it did not differ with GSE+SDF group (p = 0.12). There was a significant difference between the elastic modulus of the groups when analyzed, using one-way ANOVA (P<0.001). Tukey HSD test revealed that elastic modulus of SDF group to be significantly higher when compared to the other groups (P<0.001). The lowest mean of elastic modulus was detected in the control group (1.15±0.41), which was significantly lower than the SDF (P<0.001), GSE (P<0.001), and the CD (P = 0.02) (Fig 2). No significant difference was found between the GSE+ SDF group and the control group (P = 0.87). Elastic modulus of CD was not significantly different from GSE and GSE +SDF group (p = 0.08 and p = 0.18).

## SEM analysis

Inter-tubular dentin of CD was porous and caries crystals were seen in tubular lumens. Also bacterial cell bodies were present (Fig 3A and 3B). In the control group after pH cycling rough surface and porous structure of inter-tubular dentin was the representative of demineralization effect (Fig 3C and 3D). The organic matrix was defined as the result of mineral loss. With higher magnifications tubular wall dentin was disrupted and cracks were evident (Fig 3D). Some spaces between collagen fibers were also detected (Fig 3D). In the GSE group, irregular precipitates were seen in tubular lumen, but inter-tubular dentin was homogenous (Fig 4A and 4B). After pH cycling the precipitates were removed and inter-tubular dentin was rough with more spaces between collagen fibers (Fig 4C and 4D). In spite of degradation caused by caries, the reticular structure of organic matrix was preserved and was more dominant

**Table 1. Means ± SD of Knoop hardness and modulus of elasticity.**

| Study Groups | $H_0$ | $H_1$ | $H_1$-$H_0$ | E(Mpa) |
|---|---|---|---|---|
| CD | - | - | - | 2.22±0.81[A,C] |
| Control | 9.29±0.55[A,a] | 8.40±0.63[b] | -0.89±0.70[A] | 1.15±0.41[B] |
| GSE | 8.71±0.57[A,a] | 9.63±1.19[a] | 0.92±1.54[B] | 3.10±0.90[C] |
| SDF | 9.39±0.90[A,a] | 14.48±0.42[b] | 5.12±0.94[C] | 4.72±0.85[D] |
| GSE+SDF | 8.87±1.03[A,a] | 8.90±0.91[a] | 0.03±1.06[A,B] | 1.47±0.62[A,B] |

Abbreviations: CD, carious dentin; GSE, grape seed extract; SDF, silver diamine fluoride.

*H0, baseline hardness; H1, post-treatment hardness; E (MPa), elastic modulus (mega pascal).

**Same upper-case letters indicate no significant difference within each column. Same lower-case letters indicate no significant difference within each row.

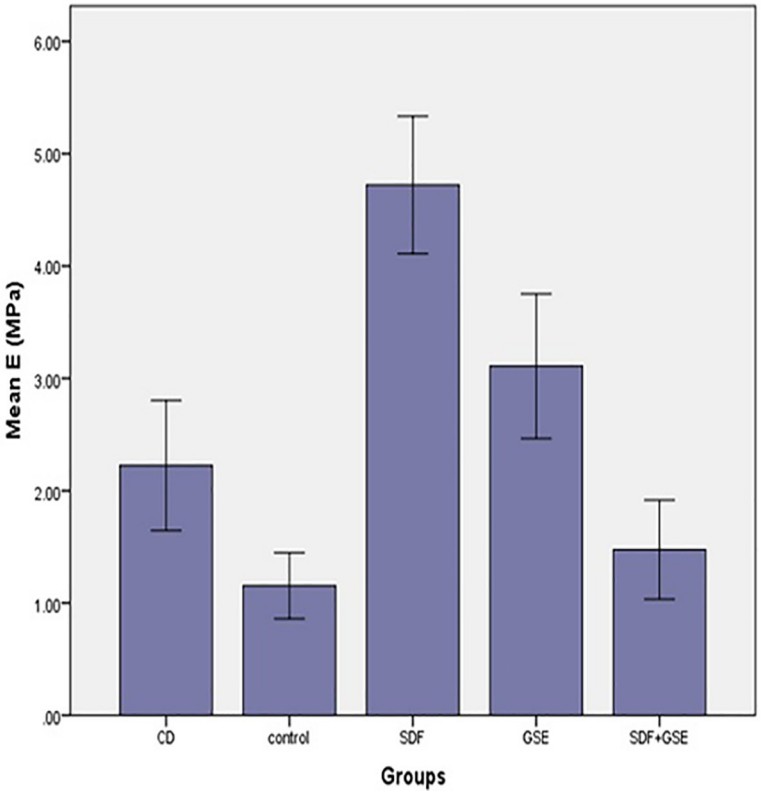

**Fig 1. Difference in hardness (H1-H0) in the study groups.**

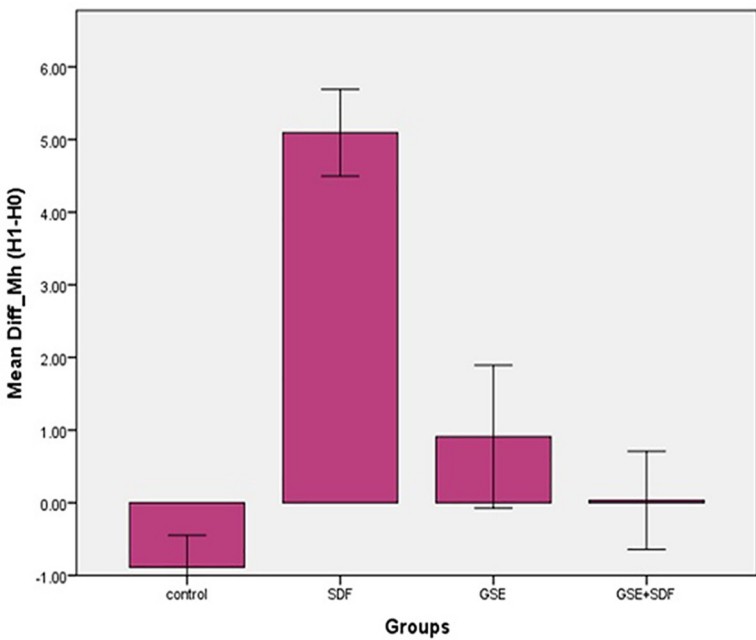

**Fig 2. Elastic modulus of the study groups.**

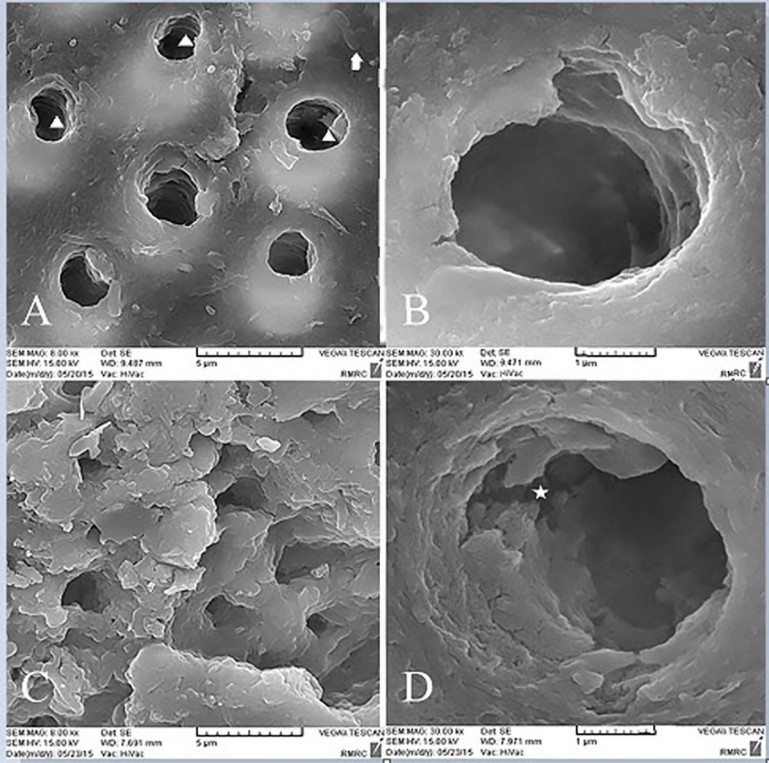

**Fig 3. SEM images of the control group.** A,B. Before acidic challenge. C,D. After Acidic challenge. Arrow shows bacterial cell bodies, triangle shows caries crystal, and star shows the crack.

compared to the control group (Fig 4C and 4D). SDF treated dentin surface was covered with a dense layer of granular deposit and the tubules were obstructed (Fig 5A and 5B). After acid challenge in this SDF group, the sediments were largely preserved, and it appeared that the surface was less cavernous than the control group, where the tubular dentin walls were intact (Fig 5C and 5D). Simultaneous treatment with GSE and SDF resulted in the formation of cubic particles that were dissociated into fine particles after pH cycling (Fig 6). Analysis of the amounts of the depositions revealed the highest value in SDF group before acid challenge and the lowest value in GSE group after acid challenge.

## Discussion

Three perquisites for dentin Biomodification are: 1. Preservation of collagen scaffold and inter-fibrillar cross-links, 2. Presence of seed mineral crystals, 3. Precipitation and penetration of mineralizing agents into underlying demineralized tissue [28]. Demineralization during the carious process leads to reduced mineral content that consequently results in diminished mechanical properties [29, 30]. Severity of a carious lesion is measured by the alterations in the mechanical properties; likewise, the best criterion in evaluating the success of tissue repair is the assessment of mechanical properties. Although the amount of mineral deposits was previously used to study the effectiveness of remineralization process, recently, the biomechanical aspects of dentin remineralization has been emphasized. This is due to the fact that the successful repair of demineralized dentin not only relies on the mineral deposits, but also heavily depends on the position of these sediments within the organic matrix. Organic-inorganic component relationship directly affects the mechanical properties of dentin. Intra fibrillary

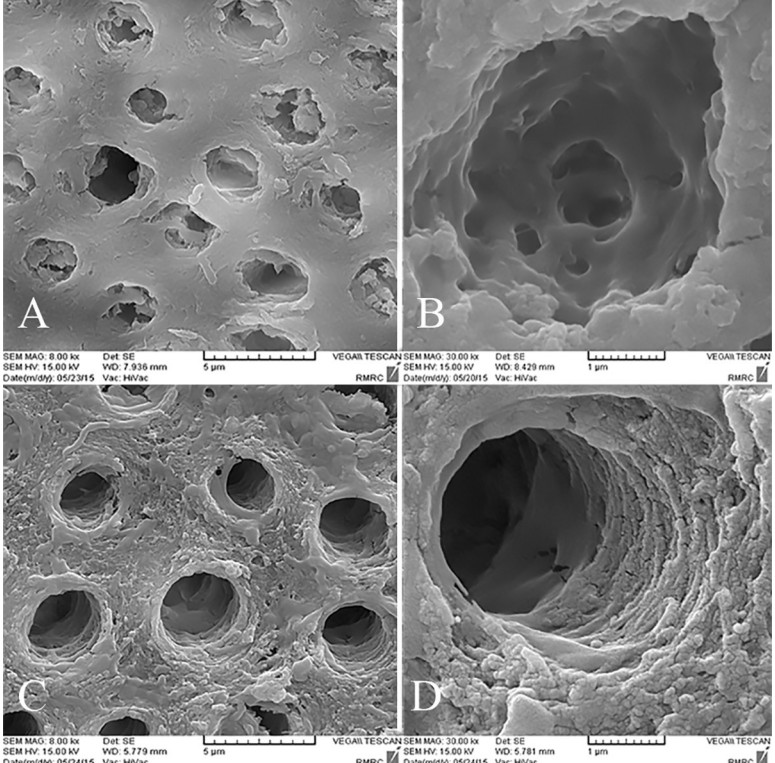

**Fig 4. SEM images of the GSE group.** A,B. Before acidic challenge. C,D. After Acidic challenge.

mineralization is very important with this regard [4, 2, 31]. Recovery of mechanical properties is a reliable way to access dentin caries arrest or repair. Deposition of minerals in the surface layer (surface mineralization) can be measured by hardness test and intrafibrillar mineral formation (functional mineralization) and preservation of collagen cross links can be accessed by elastic modulus measurement [4, 32, 33]. In conjunction with remineralizing agents, such as calcium and phosphate, bioglass and fluoride MMP inhibitors were used in an attempt to promote remineralization [27, 34, 35]. Some studies have attempted in biomimetic remineralization of dentin and showed promising results [1, 2, 31, 36]. Preservation of dentin collagen scaffold resulted in suppression of carious activities as well as prevention of demineralization cycle [37]. It has been shown that degradation of the organic matrix using collagenases increases the likelihood of acid-induced demineralization in dentin carious lesions [38].

The present study, investigated the effect of SDF, GSE, and their combination on mechanical properties and microstructure of dentin carious lesions submitted to acid challenges. The findings revealed that the carious dentin exposed to SDF showed higher microhardness and elastic modulus compared to the untreated dentin after pH cycles. GSE treatment compensated for the effects pH cycles in comparison with the control group. Therefore, the null hypothesis was partially rejected. Since GSE+SDF did not increase the resistance of carious dentin to acid challenges; thus this result approved the null hypothesis.

Our previous study showed the improvement of mechanical properties of carious dentin after SDF treatment [8]. This beneficial effect persisted after acid challenge, which can be attributed to increased mineral content. High concentration of fluoride in SDF solution is favorable in inhibiting dentin demineralization. In line with this result, a study by Chu et al. [16] revealed that the regular use of SDF for three months increased microhardness of dentin

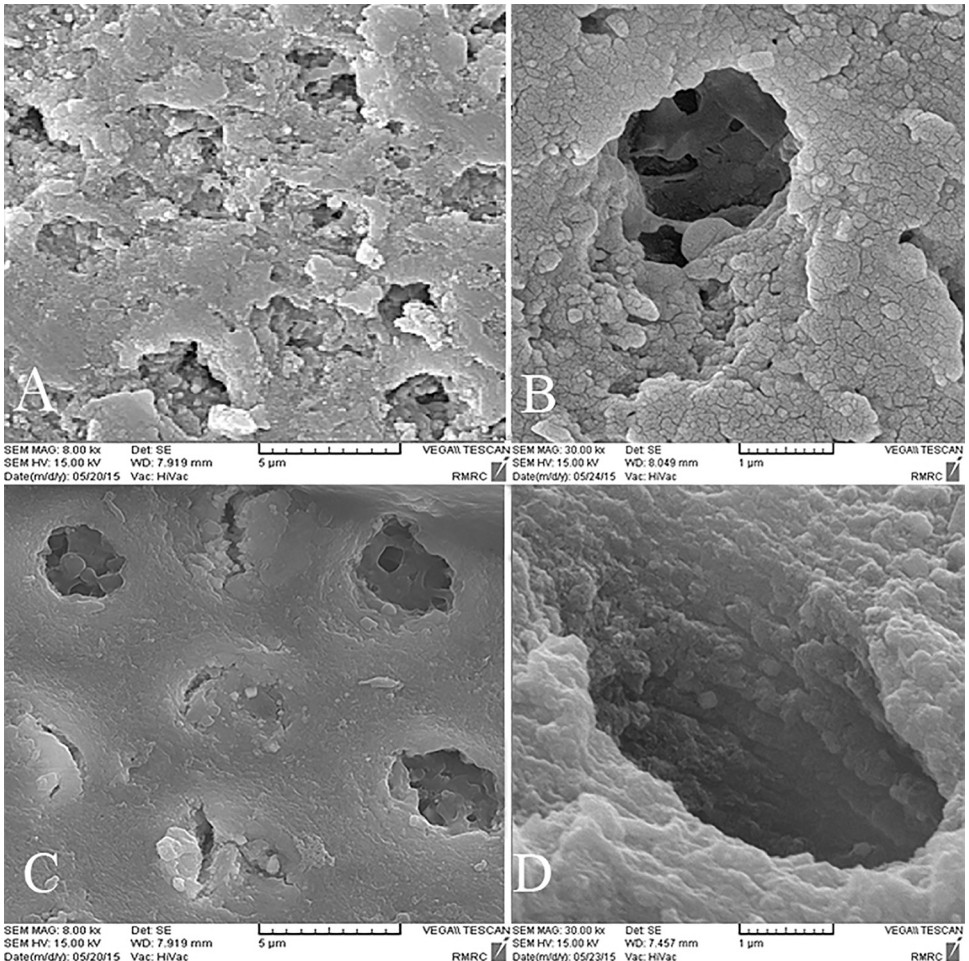

**Fig 5. SEM images of the SDF group.** A,B. Before acidic challenge. C,D. After Acidic challenge.

carious lesions. In an investigation by Mei *et al*. [17], the effects of SDF on slowing the demineralization process were proven. In another study by Mei *et al*. [39], it was found that a 24-month biannual application of SDF resulted in arrested dentin carious lesions. SDF chemically reacts with the remaining hydroxyapatite in carious dentin. The formula below shows the reaction that occurs between the two substances [40].

$$Ca_{10}(PO_4)_6(OH)_2 + Ag(NH_3)_2F \longrightarrow Ag_3PO_4 + CaF_2 + NH_4OH$$

One of the products of this reaction is $CaF_2$, which acts as a fluoride reservoir. During acid challenge it slowly releases fluoride to regulate pH and form fluorapatite, which is more acid resistant [19]. SDF is an alkaline solution with pH of 10. This condition favors the formation of fluorapatite [41]. In addition to its role in strengthening the mineral component, fluoride protects organic matrix of dentin by two possible mechanisms. Exposed collagen fibers are prone to protease activity of the enzymes. Mineral crystals can shield collagen molecule by adhering to calcium binding sites. SDF treated carious dentin exhibited less denuded collagen fibers. The second mechanism is based on the strong inhibitory effect of fluoride ion on MMP 2, MMP8, and MMP 9. Cathepsins B and K were also shown to be inhibited by fluoride. This inhibitory effect takes action in few minutes by binding to Zinc and Calcium ions, required for

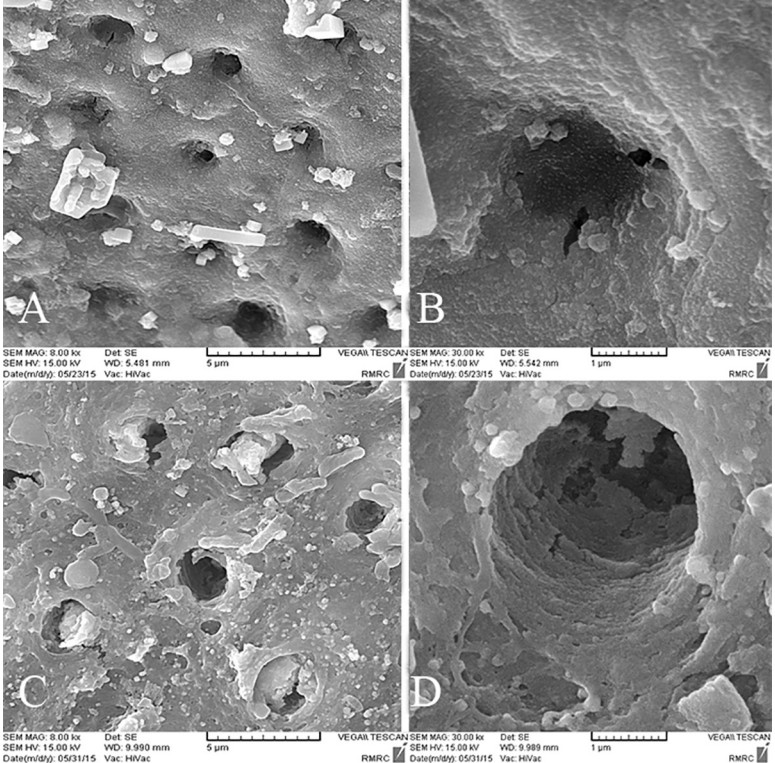

**Fig 6. SEM images of the GSE+SDF group.** A,B. Before acidic challenge. C,D. After Acidic challenge.

the activation of MMPs. Also alkaline pH of SDF can prohibit MMPs and cathepsins activation [42]. Phosphate ions form covalent bonds with collagen molecules which are facilitated under alkaline condition. Thereafter binding of calcium ions results in apatite nucleation [39]. $Ag_3PO_4$ is another major product of this reaction, which reacts with alkaline chloride solutions to form AgCl. This silver-containing compound has lower solubility than $Ag_3PO_4$ and can provide a more stable antimicrobial effect. Silver interacts with the thiol groups of bacterial DNA that inhibit cell function, and leading to bacterial cell death. This is the mechanism of cariogenic biofilm inhibition and bactericidal effects of SDF. MMPs inhibition and protection of dentin collagen matrix is also attributed to silver ion. AgCl correlates with surface hardening of the lesion [41]. Phosphate ions also participated in fluorapatite formation. High concentration of calcium, fluoride, and phosphate ions after SDF treatment might contribute in deep remineralization of carious lesions [43]. The synergistic effect of silver and fluoride ions in the inhibition of cariogenic biofilm growth, remineralization, and protection of organic matrix is the reason for SDF effectiveness to arrest the pre-exiting dentin caries and preventing new caries formation. Clustered granular dense structures formed after SDF treatment and were largely preserved after pH cycling. $CaF_2$-like globules were also detected. Collagen fibers of tubular wall dentin were also coated with this dense layer. This observation presumably explains how SDF could protect and preserve dentin organic structure.

Although there was no significant difference between $H_1$ and $H_0$ of GSE group, $H_1$-$H_0$ and elastic modulus was higher than the control group. This result can be attributed to prevention of demineralization and strengthening of organic matrix by GSE, and corroborated the previous findings [27, 44–46]. Cross-links of collagen fibers are affected by the low pH of caries process and demineralization cycles, leading to decreased elastic modulus [47]. PA from GSE is a

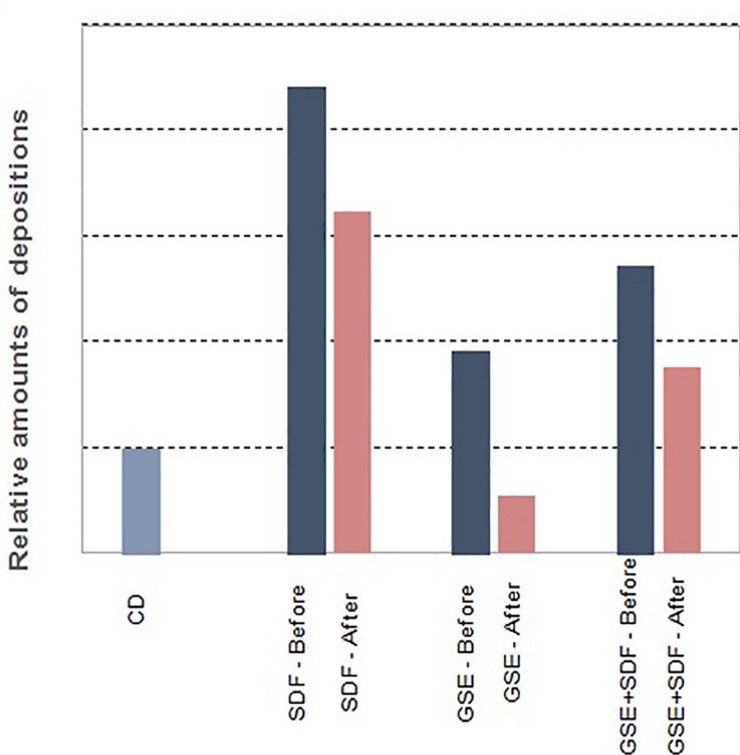

**Fig 7. Histogram of depositions in the carious dentin and experimental conditions before and after acid challenge.**

non-specific MMP inhibitor and a collagen cross-linker. Crosslinking agents stiffen the collagen molecules so that they cannot untwist or slide pass each other under mechanical forces [11]. It has been proposed that PA can bind to metallic ions *via* some functional groups. Thus, it might prevent calcium and phosphate ions chelation from dentin during acid challenge [48]. In addition, PA can bind to calcium ions from remineralization solution and promote carious dentin calcification. PAs create covalent cross-links in collagen, and furthermore they can directly interfere with the mobility and function of proteinases *via* cross-linkage and changing enzyme structure [49]. Inactivation of cysteine cathepsins B and K by PA was shown [5]. Moreover, through PA–collagen interaction, GSE might interact with the organic dentin matrix, leading to the stabilization of the exposed collagen matrix [10]. Previous studies reported that the demineralized dentin matrix treated with PA exhibited higher stiffness [25, 50–52]. However, our previous findings showed that despite the increase in hardness GSE could not increase elastic modulus of Carious dentin [8]. The application time of GSE was increased to 30 minutes in this study compared to 10 minute application in our previous study. However this duration is not clinically acceptable and shorter application times with agitation is recommended to be studied. In the present study, the results of SEM analysis confirmed the previous studies that showed a stabilized collagen network following the GSE treatment. As shown in Fig 4, despite the relative degradation of collagen network after the caries process, there was a more intact collagen matrix after the acid challenge in the GSE group in comparison with the other groups.

SDF and GSE showed no synergistic effect in improving the mechanical properties of dentin carious lesions. No significant difference was found between the initial and final hardness

in the GSE + SDF group. There was no significant difference between the $H_1$-$H_0$ value of the GSE + SDF and the control group. Although it has been suggested that cross-linked collagen fibers provide more space for subsurface diffusion of minerals [53], in our previous study we concluded that GSE + SDF could not strengthen carious dentin, which might be attributed to the chemical interaction of the GSE solution with SDF [8]. This interaction leads to loss of efficacy of both solutions. Removal of excess GSE solution before SDF treatment might alleviate the reaction. It has been found that hesperidin and GSE promoted dentin remineralization [10, 53, 54]. Unlike the fast process of remineralization by SDF in this study, they used remineralization cycles. These cycles allowed for gradual mineral diffusion into the lesion. Two previous studies proposed a synergistic effect between PA with casein phosphopeptide amorphous calcium phosphate and tri-calcium phosphate. In both of those studies PA solution was mixed with mineralizing agents and the test solution was applied in each cycle [27, 46]. Cai *et al*. [28] reported that the combined treatment of PA and SDF/KI resulted in a more consistent mineral distribution throughout the lesions, leading to a more significant increase in surface and cross-sectional microhardness and elastic modulus than SDF/KI alone. This controversial result could be due to the application of KI and the chemical model used by Cai *et al*. which consisted of only demineralization cycles. As for the acid challenge regimens used in this study, natural caries process comprises of both demineralization and remineralization cycles. In the aforementioned studies, GSE powder was used with different protocols and was from different sources. Previous studies found that the results obtained with PA were strongly dependent on its origin, the solvent used for its extraction, concentration, and exposure time [5]. The acidic pH of the solution was neutralized in the present study but Cai *et al*. used the PA solution with pH of 4.46. Formation of cubic particles after GSE+SDF treatment might corroborate the proposed chemical interaction of the two agents. However in the study by Cai *et al*. cubic particles were seen after SDF treatment which was different from the observations of this study and previous studies [6, 39]. The evidence indicate that polyphenolic compounds available in plant extracts are potential reducing agents for silver ions and can be used in the production of nano silver particles *via* a process known as green synthesis [55]. GSE was successfully used for this purpose [56, 57], but these nanoparticles are extremely unstable and must be capped with appropriate capping agents to provide stability and prevent agglomeration. Otherwise, silver would go through a reaction to form silver compounds, such as $Ag_2O$. Rapid color change from light brown to dark brown after the application of SDF on GSE treated dentin surface can be related to the formation of $Ag_2O$. Also, if sulfide ligands are available in the composition of GSE, formation of $Ag_2S$ is possible. Therefore, $Ag_2O$, $Ag_2S$ or a mixture of these agents might be the constituents of the cubic particles. The solutions used in pH cycling protocol contained $CaCl_2$ and $KH_2PO_4$. Considering the complexation constants formation of $Ag_3PO_4$ and AgCl is also possible after PH cycling. This can explain the changes in the crystalline structure. Other possible explanation is that the pH cycling might merely change the morphology of the crystals without any alteration to their composition.

The largest amount of depositions was observed in SDF group (Fig 7) and this was line with hardness and elastic modulus values and can be explained as functional mineralization. However the amount of depositions and mechanical properties were not concordant for GSE+SDF group. This finding can be attributed to the superficial deposits rather than functional mineralization.

To simulate variations in mineral saturation and pH in the natural caries process, the *in-vitro* pH-cycling models were introduced [58]. In the present study, the pH-cycling model was used to simulate cariogenic status of the oral environment for the development of caries. In this model, the dentin carious lesions were exposed to repeated demineralization and remineralization cycles. Salivary flow provides the essential ions during caries remineralization cycles

in the oral environment. Calcium and phosphate ions are especially important, since both are the structural components of hydroxyapatite unit cell. $CaCl_2$ and $KH_2PO_4$ in remineralization solution were a basic simulation of this salivary environment. The residual mineral crystals of the tooth could be another important factor of remineralization. These crystals act as nucleating sites [41]. Given that the dentin carious lesions varies in nature, and developmental stage the insignificant difference in initial hardness of the specimens implies that the selection criteria was able to control the inter-specimen variations. Tensile test used to measure elastic modulus is a destructive test. Therefore the baseline and final elastic modulus could not be measure on the same specimen. The present study investigated the mechanical and moicrostructural effects of the test solutions. The strong point of this study is the macroscale evaluation of mechanical properties which has closer relation with clinical conditions. Increased dentin hardness, especially in root carious lesions, reduces wear and abrasion and an increase in the elastic modulus results in reduced deflection in the cervical region. SDF is considered as a potential agent in nonrestorative cavity control treatments of root caries due to its proven antibacterial properties [17] as well as the successful remineralization and regeneration of dentin mechanical properties. However it is recommended that the biochemical aspects be studied in future. One limitation of the current was to use coronal dentin instead of root dentin. This was due to the difficulties in preparing specimens for tensile test from root dentin. Acid challenge protocol used in this *in vitro* study was a partial simulation of the cariogenic cycles in the oral environment. The role of bacteria and proteolytic enzymes should not be overlooked. Longer duration of pH cycling regimen might be beneficial in disclosing the degenerative role of activated MMPs in carious dentin tissue and the effect of test solutions in arresting dentin caries.

## Conclusions

The use of SDF on carious dentin prior to acid challenge resulted in increased mechanical properties. GSE also increased the resistance of carious dentin to pH cycling to a lesser extent than SDF. The simultaneous use of GSE and SDF prevented from reduction of hardness following pH cycling, but did not improve the modulus of elasticity.

## Supporting information

**S1 File.**
(SAV)

**S2 File.**
(SAV)

## Acknowledgments

The authors wish to thank Mr. H. Argasi at the Research Consultation Center (RCC) of Shiraz University of Medical Sciences for his invaluable assistance in editing this manuscript and Dr. Vossoughi/Sayyadi from the Center for Research Improvement of the School of Dentistry for statistical analysis.

## Author Contributions

**Conceptualization:** Maryam Firouzmandi.

**Data curation:** Hadis Sadeghi.

**Formal analysis:** Hadis Sadeghi.

**Methodology:** Fateme Vasei, Rashin Giti.

**Resources:** Fateme Vasei.

**Supervision:** Maryam Firouzmandi.

**Validation:** Rashin Giti.

**Visualization:** Rashin Giti.

**Writing – original draft:** Fateme Vasei, Hadis Sadeghi.

**Writing – review & editing:** Maryam Firouzmandi.

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
