## [Decision Letter · Decision Letter 0]

12 May 2020

PONE-D-20-09551

Effect of silver diamine fluoride and proanthocyanidin on resistance of carious dentin to acid challenges

PLOS ONE

Dear Dr. Vasei,

Thank you for submitting your manuscript to PLOS ONE. After careful consideration, we feel that it has merit but does not fully meet PLOS ONE’s publication criteria as it currently stands. Therefore, we invite you to submit a revised version of the manuscript that addresses the points raised during the review process.

Since you think that the SDF treatment can improve the mechanic properties of carious dentin and attributed it to increased mineral content. You only provide SEM images, which are only qualitative analysis. Two independent reviewers suggest you also use other direct evidence such as TEM image. 

Please also think of presenting your data of resistant to acid such by the amount of granular deposit between control, GSE group, SDF group and GSE+SDF group in histograms.

We would appreciate receiving your revised manuscript within three weeks. To enhance the reproducibility of your results, we recommend that if applicable you deposit your laboratory protocols in protocols.io, where a protocol can be assigned its own identifier (DOI) such that it can be cited independently in the future. For instructions see: http://journals.plos.org/plosone/s/submission-guidelines#loc-laboratory-protocols

We look forward to receiving your revised manuscript.

Kind regards,

Jinhui Tao, Ph.D.

Academic Editor

PLOS ONE

Journal Requirements:

1. Please provide additional information about the samples/tissue used in your study. Specifically, please ensure that you have discussed whether all samples were fully anonymized before you accessed them or whether the IRB or ethics committee waived the requirement for informed consent. If patients provided informed written consent to have their samples used in research, please include this information in the methods section.

Reviewers' comments:

Reviewer's Responses to Questions

**Comments to the Author**

1. Is the manuscript technically sound, and do the data support the conclusions?

Reviewer #1: Partly

Reviewer #2: Partly

Reviewer #3: Partly

2. Has the statistical analysis been performed appropriately and rigorously? 

Reviewer #1: Yes

Reviewer #2: Yes

Reviewer #3: Yes

3. Have the authors made all data underlying the findings in their manuscript fully available?

Reviewer #1: Yes

Reviewer #2: No

Reviewer #3: Yes

4. Is the manuscript presented in an intelligible fashion and written in standard English?

Reviewer #1: Yes

Reviewer #2: Yes

Reviewer #3: Yes

5. Review Comments to the Author

Reviewer #1: The work by Fateme et al. demonstrated that the mechanical properties of dental caries could be significant improvement by using Silver diamine fluoride (SDF) modification. It also investigated the microstructure of the dentin after acid challenges. However, the authors have not deeply researched the mechanism of the SDF or GSE on how to protect the dentin against acid. Meanwhile, many statements and conclusions the authors raised have no direct evidence to support. In a word, this work is suitable for publication in PLOS ONE, but need to provide more experimental data as evident and addressing the following concerns:

1. Missing a unit after “0.5 ± 0.1”, please see Line 123.

2. Does there exist an interaction between dentin and PA from GSE or SDF? If YES, what is the interaction between them? You can examine it through spectroscopic technology such as FTIR.

3. What is the cubic particles shown in Fig 4, which is important to understand the role of GSE and SDF?

4. The authors think that the SDF treatment can improve the mechanic properties of carious dentin and attributed it to increased mineral content. Here the authors just provide SEM images, which only qualitative analysis, and no other direct evidence such as TEM image. For TEM image can refer the following references.

[1] Wang, J. et al. Remineralization of dentin collagen by meta-stabilized amorphous calcium phosphate. CrystEngComm 15, 6151-6158, doi:10.1039/C3CE40449H (2013).

[2] Sun, J. et al. Biomimetic promotion of dentin remineralization using l-glutamic acid: inspiration from biomineralization proteins. J. Mater. Chem. B 2, 4544-4553, doi:10.1039/C4TB00451E (2014).

5. The reaction between hydroxyapatite and SDF is inaccurate, which included the formula of hydroxyapatite (Ca10(PO4)6(OH)2) and the stoichiometric numbers. Please correct the reaction, See Line 258.

6. The sentence “It has been proposed that PA can bind to metallic ions via some functional groups.” should be added a reference to support it.

7. The style of some of the list of references is incorrect, like reference #15, #16 and so on. Please according to the journal criterion to correct it.

Reviewer #2: This manuscript addresses inhibition of carious dentin via silver diamine fluoride and proanthocyanidin. The study has some strength including different methods to verify the treatment effect, but major concerns exist. See the below points to address.

• Page 6 line 134: In your experiment, specimens were treated 30 mins in GSE groups, however, specimens were treated 4 mins in SDF groups. It not clear to me why the treatment time is different? Please rectify.

• Page 10 line 205: The statement “bacterial cell bodies were present” in fig 1a, 1b. I actually not looking at the bacterial cell bodies, just the dentine in the SEM. Please verify and make an obvious mark.

• Page 10 line 208: It is unclear based on the SEM shown that resistant capacity to demineralization was assessed. I think SEM only used to observe the surface demineralization of carious dentin. However, this study requires additional experiments to confirm your conclusion. For example, observation of internal demineralization of dentin slices using TEM.

• Fig2a: Why the magnification of SEM images in Fig2a (3x) is different from Fig1a, Fig3a and Fig4a (5x)? Please provide your rationale for your choice.

• All SEM figures: Please provide more statistical analysis of resistant capacity to demineralization such as the amount of granular deposit between control, GSE group, SDF group and GSE+SDF group.

Reviewer #3: This results for SDF and GSE groups are not surprising, and SDF and GSE showed no synergistic effect. However, their conclusions look like being supported the data. Before accepting, the authors have to address the following problems:

1. Figure 1, it’s better to point out the caries crystals, the cell bodies and the cracks.

2. What’s the concentration of PA in grape seed extract?

3. Please provide details for “Some spaces between collagen fibers were also detected.”

4. Figure 2, how to make the conclusion that “the reticular structure of organic matrix was preserved and was more dominant compared to the control group”?

5. Figure 4, how to tell that the fined particles after pH cycling were from the cubic particles?

6. The writing can be improved. For example, page 8, line 166, “the mean value was established as 166 the KHN” is not a complete sentence. Page 9, line 183, “in contrast to the control group” might be better than “except the control group”. Page 10, line 218, “in the formation of cubic particles” might be better than “in formation of cubic particles”.

6. PLOS authors have the option to publish the peer review history of their article (what does this mean?). If published, this will include your full peer review and any attached files.

Reviewer #1: No

Reviewer #2: No

Reviewer #3: No

---

## [Author Response · Author response to Decision Letter 0]

23 Jun 2020

Dear editor-in-chief, plos one journal

I would like to thank you for taking into account the review of our article. The invaluable comments of the esteemed reviewers are placed within the text of the manuscript and the answers of the authors are attached in a separate document. 

I would greatly appreciate the valuable suggestion of the reviewers in order to improve the quality of the article. However, in the current situation and because of this corona pandemic, it will take a long and time-consuming legal and ethical process to obtain the necessary credentials from the university with regards to collecting the extracted teeth, given that the experiments related to this study were completed a year ago.

On the other hand, due to the current economic sanctions and severe restrictions on imports, the osmium tetroxide does not currently exist in the Iranian market. Therefore, research centers providing TEM services face extreme resource constraints and offer the test service subject to the provision of the necessary materials by the researchers themselves. Moreover, another research center was unable to provide service due to the fracture of the blade of the ultramicrotom device. Therefore, considering the financial, timely and legal burden of this issue, I would like to kindly ask you to reconsider the requirement of providing TEM images.

Finally, I would like to point out that although the amount of mineral deposits was previously used to study the effectiveness of remineralization process, recently the biomechanical aspects of dentin remineralization has been emphasized. This is due to the fact that the successful repair of demineralized dentin not only relies on the mineral deposits, but also heavily depends on the position of these sediments within the organic matrix. Organic-inorganic component relationship directly affects the mechanical properties of dentin. Intra fibrillar mineralization is very important with this regard. The aim of carious dentin biomodification is to recover its functionality. Severity of a carious lesion is measured by the alterations in the mechanical properties; likewise the best criterion in evaluating the success of tissue repair is the assessment of mechanical properties. Similarly, in a study by Chen et al., micromechanical properties have been studied to confirm the findings of ultrastructural analyses. In our study, superficial remineralization was examined by means of the hardness test and functional remineralization by tensile test. The tensile test results can be related to the properties of the bulk of the material. The strong point of this study is the macroscale evaluation of mechanical properties which has closer relation with clinical conditions. Increased dentin hardness, especially in root carious lesions, reduces wear and abrasion and an increase in the elastic modulus results in reduced deflection in the cervical region. Finally, these properties will result in the restoration of tissue functionality. SDF can be considered as a potential agent in nonrestorative cavity control treatments of root caries due to its proven antibacterial properties as well as the successful remineralization and regeneration of dentin mechanical properties.

A histogram of depositions is provided. Also surface plot for control and GSE+SDF group is prepared. If you think that it is useful we can prepare the plot for other groups.

Corresponding author

Fateme vasei

Dear Academic Editor:

I would greatly appreciate your valuable suggestion in order to improve the quality of the article. However, in the current situation and because of this corona pandemic, it will take a long and time-consuming legal and ethical process to obtain the necessary credentials from the university with regards to collecting the extracted teeth, given that the experiments related to this study were completed a year ago.

On the other hand, due to the current economic sanctions and severe restrictions on imports, the osmium tetroxide does not currently exist in the Iranian market. Therefore, research centers providing TEM services face extreme resource constraints and offer the test service subject to the provision of the necessary materials by the researchers themselves. Moreover, another research center was unable to provide service due to the fracture of the blade of the ultramicrotom device. Therefore, considering the financial, timely and legal burden of this issue, I would like to kindly ask you to reconsider the requirement of providing TEM images.

Finally, I would like to point out that although the amount of mineral deposits was previously used to study the effectiveness of remineralization process, recently the biomechanical aspects of dentin remineralization has been emphasized. This is due to the fact that the successful repair of demineralized dentin not only relies on the mineral deposits, but also heavily depends on the position of these sediments within the organic matrix. Organic-inorganic component relationship directly affects the mechanical properties of dentin. Intra fibrillar mineralization is very important with this regard. The aim of carious dentin biomodification is to recover its functionality. Severity of a carious lesion is measured by the alterations in the mechanical properties; likewise the best criterion in evaluating the success of tissue repair is the assessment of mechanical properties. Similarly, in a study by Chen et al., micromechanical properties have been studied to confirm the findings of ultrastructural analyses. In our study, superficial remineralization was examined by means of the hardness test and functional remineralization by tensile test. The tensile test results can be related to the properties of the bulk of the material. The strong point of this study is the macroscale evaluation of mechanical properties which has closer relation with clinical conditions. Increased dentin hardness, especially in root carious lesions, reduces wear and abrasion and an increase in the elastic modulus results in reduced deflection in the cervical region. Finally, these properties will result in the restoration of tissue functionality. SDF can be considered as a potential agent in nonrestorative cavity control treatments of root caries due to its proven antibacterial properties as well as the successful remineralization and regeneration of dentin mechanical properties.

A histogram of depositions is provided. Also surface plot for control and GSE+SDF group is prepared as below. If you think that it is useful we can prepare the plot for other groups.

Surface Plot of Control

Surface Plot of GSE+SDF-before

Reviewer #1:

1. Missing a unit after “0.5 ± 0.1”, please see Line 123.

 The missing unit (mm) is added.

2. Does there exist an interaction between dentin and PA from GSE or SDF? If YES, what is the interaction between them? You can examine it through spectroscopic technology such as FTIR. 

 Interaction of SDF with tooth components was investigated in a previous study (Lou YL, Botelho MG, Darvell BW. 2011. Reaction of silver diamine [corrected] fluoride with hydroxyapatite and protein. J Dent. 39(9):612–618.). Some points about the reaction mechanism were explained in the discussion of the manuscript. Different interaction mechanisms are proposed between dentin and PA: covalent interaction, ionic interaction, hydrogen bonding. One of the accepted mechanisms is the formation of calcium ion bridges. PA catechol moieties cross-link within collagen via the formation of hydrogen and covalent bonds with -NH2,-COOH, -OH, -SH functional groups (Al-Ammar A, Drummond JL, Bedran-Russo AK. The use of collagen cross-linking agents to enhance dentin bond strength. J Biomed Mater Res B Appl Biomater 2009; 91: 419-424.)( Biostability of the Proanthocyanidins-Dentin Complex and Adhesion Studies A.A. Leme-Kraus1, B. Aydin1, C.M.P. Vidal1, R.M Phansalkar2, J.W. Nam2,J. McAlpine2, G.F. Pauli2, S. Chen2, and A.K. Bedran-Russo1.) Since the reaction mechanisms were investigated previously this study focused on mechanical properties.

3. What is the cubic particles shown in Fig 4, which is important to understand the role of GSE and SDF?

There are evidences indicating that polyphenolic compounds available in plant extracts are potential reducing agents for silver ions and can be used in production of nano silver particles via a process known as green synthesis (Biosynthesis of silver nanoparticles with adiantumcapillus-veneris L leaf extract in the batch process and assessment of antibacterial activitySariyeh Omidi, Sajjad Sedaghat, Kambiz Tahvildari, Pirouz Derakhshi & Fereshte Motiee.) GSE was successfully used for this purpose (‘Green’ Synthesis of Silver Nanoparticles by Using Grape (Vitis vinifera) Fruit Extract: Characterization of the Particles and Study of Antibacterial Activity Kaushik Roy* ¹, Supratim Biswas², and Pataki C Banerjee ¹)(Green synthesis of silver nanoparticles using grape seed extract and their application for reductive catalysis of Direct Orange. panelPingYaoacJunZhangcTielingXingabGuoqiangChena RanTaocKwang-HoChoo) .

 However these nano particles are extremely unstable and must be capped with suitable capping agents to provide stability and prevent agglomeration. Otherwise silver would go through a reaction to form silver compounds such as Ag2O or other Ag complexes with ligands which are in the solution. Rapid color change from light brown to dark brown after application of SDF on GSE treated dentin surface can be probably related to the formation of Ag2O or mentioned complexes. Also if sulfide ligands are available in the composition of GSE formation of Ag2S is possible. Therefore Ag2O,Ag2S or a mixture of these particles might be the constituents of the cubic particles. The solutions used in pH cycling protocol contained CaCl2 and KH2PO4. Considering the complex formation constants of Ag3PO4 and AgCl, presence of these compounds are also possible after PH cycling. This can explain the changes in the crystalline structure. Other possible explanation is that the pH cycling might only simply change the morphology of the crystals without any alteration their composition

4. The authors think that the SDF treatment can improve the mechanic properties of carious dentin and attributed it to increased mineral content. Here the authors just provide SEM images, which only qualitative analysis, and no other direct evidence such as TEM image. For TEM image can refer the following references.

[1] Wang, J. et al. Remineralization of dentin collagen by meta-stabilized amorphous calcium phosphate. CrystEngComm 15, 6151-6158, doi:10.1039/C3CE40449H (2013).

[2] Sun, J. et al. Biomimetic promotion of dentin remineralization using l-glutamic acid: inspiration from biomineralization proteins. J. Mater. Chem. B 2, 4544-4553, doi:10.1039/C4TB00451E (2014).

I would greatly appreciate the valuable suggestion of the reviewers in order to improve the quality of the manuscript and the suggested references contained useful information and I cited them in the manuscript. However, in the current situation and because of this corona pandemic, it will take a long and time-consuming legal and ethical process to obtain the necessary credentials from the university with regards to collecting the extracted teeth, given that the experiments related to this study were completed a year ago.

I would like to point out that in our study; superficial remineralization was examined by means of the hardness test and functional remineralization by tensile test. The tensile test results can be related to the properties of the bulk of the material. The strong point of this study is the macroscale evaluation of mechanical properties which has closer relation with clinical conditions. Increased dentin hardness, especially in root carious lesions, reduces wear and abrasion and an increase in the elastic modulus results in reduced deflection in the cervical region.

5. The reaction between hydroxyapatite and SDF is inaccurate, which included the formula of hydroxyapatite (Ca10(PO4)6(OH)2) and the stoichiometric numbers. Please correct the reaction, See Line 258.

The reaction formula is corrected as being: 

 Ca10 (PO4)6 (OH)2+Ag (NH3)2F Ag3PO4 +CaF2 +NH4OH

However the formula is unbalanced and stoichiometric numbers are not considered. A reference (Reaction of silver diamine fluoride with hydroxyapatite and protein Y.L. Lou a, M.G. Botelho a,*, B.W. Darvell b) is also added.

6. The sentence “It has been proposed that PA can bind to metallic ions via some functional groups.” should be added a reference to support it.

Reference "Tsao R. Chemistry and biochemistry of dietary polyphenols. Nutrients 2010; 2: 1231-1246." is added. 

7. The style of some of the list of references is incorrect, like reference #15, #16 and so on. Please according to the journal criterion to correct it.

Correction is made.

Reviewer #2:

• Page 6 line 134: In your experiment, specimens were treated 30 mins in GSE groups, however, specimens were treated 4 mins in SDF groups. It not clear to me why the treatment time is different? Please rectify. 

According to " Castellan CS, Pereira PN, Viana G, Chen SN, Pauli GF, Bedran-Russo AK. Solubility study of phytochemical cross-linking agents on dentin stiffness. J Dent 2010; 38: 431-436." The elastic modulus of dentin was significantly increased by the PA treatment regardless of time. They tested 10, 30, 60, 120 and 240 min. the results of our previous study showed that 10 min application of GSE could not increase elastic modulus of inner carious dentin though we extended the application time to 30 min to investigate the effect of time. Application time of 1-4 min for SDF was recommended by the manufacturer and was also applied by previous studies for example " Quock RL, Barros JA, Yang SW, Patel SA. Effect of silver diamine fluoride on microtensile bond strength to dentin. Oper Dent 2012; 37: 610-616."

• Page 10 line 205: The statement “bacterial cell bodies were present” in fig 1a, 1b. I actually not looking at the bacterial cell bodies, just the dentine in the SEM. Please verify and make an obvious mark.

 Bacterial cell bodies are defined with arrow.

• Page 10 line 208: It is unclear based on the SEM shown that resistant capacity to demineralization was assessed. I think SEM only used to observe the surface demineralization of carious dentin. However, this study requires additional experiments to confirm your conclusion. For example, observation of internal demineralization of dentin slices using TEM.

I would greatly appreciate the valuable suggestion of the reviewers in order to improve the quality of the manuscript. However, in the current situation and because of this corona pandemic, it will take a long and time-consuming legal and ethical process to obtain the necessary credentials from the university with regards to collecting the extracted teeth, given that the experiments related to this study were completed a year ago.

I would like to point out that in our study; superficial remineralization was examined by means of the hardness test and functional remineralization by tensile test. The tensile test results can be related to the properties of the bulk of the material. The strong point of this study is the macroscale evaluation of mechanical properties which has closer relation with clinical conditions. Increased dentin hardness, especially in root carious lesions, reduces wear and abrasion and an increase in the elastic modulus results in reduced deflection in the cervical region.

• Fig2a: Why the magnification of SEM images in Fig2a (3x) is different from Fig1a, Fig3a and Fig4a (5x)? Please provide your rationale for your choice.

Fig2a is replaced with the image with correct magnification.

• All SEM figures: Please provide more statistical analysis of resistant capacity to demineralization such as the amount of granular deposit between control, GSE group, SDF group and GSE+SDF group.

/A histogram of depositions is provided. Also surface plot for control and GSE+SDF group is prepared as below. If you think that it is useful we can prepare the plot for other groups.

 Surface Plot of Control 

Surface Plot of GSE+SDF-before

Reviewer #3:

1. Figure 1, it’s better to point out the caries crystals, the cell bodies and the cracks.

All done.

2. What’s the concentration of PA in grape seed extract?

 The concentration was >90% as stated by the manufacturer.

3. Please provide details for “Some spaces between collagen fibers were also detected.”

This sentence was referred to the fig 1-d.when compared with fig 1-b more porous structure of inter-tubular and intra-tubule dentin was seen.

4. Figure 2, how to make the conclusion that “the reticular structure of organic matrix was preserved and was more dominant compared to the control group”?

 After pH cycling the dentin surface was disrupted and eroded in the control group (fig 1c), but in the GSE group following pH cycling despite porous dentin surface the consistency of dentin matrix was preserved.

5. Figure 4, how to tell that the fined particles after pH cycling were from the cubic particles?

There are evidences indicating that polyphenolic compounds available in plant extracts are potential reducing agents for silver ions and can be used in production of nano silver particles via a process known as green synthesis (Biosynthesis of silver nanoparticles with adiantumcapillus-veneris L leaf extract in the batch process and assessment of antibacterial activitySariyeh Omidi, Sajjad Sedaghat, Kambiz Tahvildari, Pirouz Derakhshi & Fereshte Motiee.) GSE was successfully used for this purpose (‘Green’ Synthesis of Silver Nanoparticles by Using Grape (Vitis vinifera) Fruit Extract: Characterization of the Particles and Study of Antibacterial Activity Kaushik Roy* ¹, Supratim Biswas², and Pataki C Banerjee ¹)(Green synthesis of silver nanoparticles using grape seed extract and their application for reductive catalysis of Direct Orange. panelPingYaoacJunZhangcTielingXingabGuoqiangChena RanTaocKwang-HoChoo) .

 However these nano particles are extremely unstable and must be capped with suitable capping agents to provide stability and prevent agglomeration. Otherwise silver would go through a reaction to form silver compounds such as Ag2O or other Ag complexes with ligands which are in the solution. Rapid color change from light brown to dark brown after application of SDF on GSE treated dentin surface can be probably related to the formation of Ag2O or mentioned complexes. Also if sulfide ligands are available in the composition of GSE formation of Ag2S is possible. Therefore Ag2O,Ag2S or a mixture of these particles might be the constituents of the cubic particles. The solutions used in pH cycling protocol contained CaCl2 and KH2PO4. Considering the complex formation constants of Ag3PO4 and AgCl, presence of these compounds are also possible after PH cycling. This can explain the changes in the crystalline structure. Other possible explanation is that the pH cycling might only simply change the morphology of the crystals without any alteration their composition

6. The writing can be improved. For example, page 8, line 166, “the mean value was established as 166 the KHN” is not a complete sentence. Page 9, line 183, “in contrast to the control group” might be better than “except the control group”. Page 10, line 218, “in the formation of cubic particles” might be better than “in formation of cubic particles”.

The manuscript has been edited and grammatically revised.

---

## [Decision Letter · Decision Letter 1]

20 Aug 2020

Effect of silver diamine fluoride and proanthocyanidin on resistance of carious dentin to acid challenges

PONE-D-20-09551R1

Dear Dr. Vasei,

We’re pleased to inform you that your manuscript has been judged scientifically suitable for publication and will be formally accepted for publication once it meets all outstanding technical requirements.

Kind regards,

Jinhui Tao, Ph.D.

Academic Editor

PLOS ONE

Additional Editor Comments (optional):

Reviewers' comments:

Reviewer's Responses to Questions

**Comments to the Author**

1. If the authors have adequately addressed your comments raised in a previous round of review and you feel that this manuscript is now acceptable for publication, you may indicate that here to bypass the “Comments to the Author” section, enter your conflict of interest statement in the “Confidential to Editor” section, and submit your "Accept" recommendation.

Reviewer #1: (No Response)

Reviewer #2: All comments have been addressed

Reviewer #3: All comments have been addressed

2. Is the manuscript technically sound, and do the data support the conclusions?

Reviewer #1: (No Response)

Reviewer #2: Yes

Reviewer #3: Yes

3. Has the statistical analysis been performed appropriately and rigorously? 

Reviewer #1: (No Response)

Reviewer #2: Yes

Reviewer #3: Yes

4. Have the authors made all data underlying the findings in their manuscript fully available?

Reviewer #1: (No Response)

Reviewer #2: Yes

Reviewer #3: (No Response)

5. Is the manuscript presented in an intelligible fashion and written in standard English?

Reviewer #1: (No Response)

Reviewer #2: Yes

Reviewer #3: Yes

6. Review Comments to the Author

Reviewer #1: (No Response)

Reviewer #2: Author should pay attention to the author conflicts, research ethics, or publication ethics and so on.

Reviewer #3: I agree that their conclusion is supported by the data. However, figures 1 and 2 were not cited in the manuscript.

7. PLOS authors have the option to publish the peer review history of their article (what does this mean?). If published, this will include your full peer review and any attached files.

Reviewer #1: No

Reviewer #2: No

Reviewer #3: No

---

## [Editor Report · Acceptance letter]

27 Aug 2020

PONE-D-20-09551R1 

Effect of silver diamine fluoride and proanthocyanidin on resistance of carious dentin to acid challenges 

Dear Dr. Vasei:

I'm pleased to inform you that your manuscript has been deemed suitable for publication in PLOS ONE. Congratulations! Your manuscript is now with our production department. 

Kind regards, 

on behalf of

Dr. Jinhui Tao 

Academic Editor

PLOS ONE